# ChemistryQA: A Complex Question Answering Dataset from Chemistry

## Abstract

Many Question Answering (QA) tasks have been studied in NLP and employed to evaluate the progress of machine intelligence. One kind of QA tasks, such as Machine Reading Comprehension QA, is well solved by end-to-end neural networks; another kind of QA tasks, such as Knowledge Base QA, needs to be translated to a formatted representations and then solved by a well-designed solver. We notice that some real-world QA tasks are more complex, which cannot be solved by end-to-end neural networks or translated to any kind of formal representations. To further stimulate the research of QA and development of QA techniques, in this work, we create a new and complex QA dataset, ChemistryQA, based on real-world chemical calculation questions. To answer chemical questions, machines need to understand questions, apply chemistry and math knowledge, and do calculation and reasoning. To help researchers ramp up, we build two baselines: the first one is BERT-based sequence to sequence model, and the second one is an extraction system plus a graph search based solver. These two methods achieved 0.164 and 0.169 accuracy on the development set, respectively, which clearly demonstrates that new techniques are needed for complex QA tasks. ChemistryQA dataset will be available for public download once the paper is published.

## 1    Introduction

Recent years have witnessed huge advances for the question answering (QA) task, and some AI agents even beat human beings. For example, IBM Watson won Jeopardy for answering questions which requires a broad range of knowledge (Ferrucci, 2012). Transformer-based neural models, e.g. XLNet (Yang et al., 2019) and RoBERTa (Liu et al., 2019), beat human beings on both machine reading comprehension and conversational QA task. Ariso System (Clark et al., 2019) gets an 'Ace' for an eighth-grade science examination and is able to give 80 percent correct answers for 12th-grade science test.

Most solutions of the QA task fall into two categories, end-to-end solution and parsing plus execution. The former predicts answers with an end-to-end neural network, e.g., Reading comprehension QA (Rajpurkar et al., 2016; 2018; Lai et al., 2017) and Science Exam QA (Clark et al., 2019; 2018). The latter translates a question into a specific structural form which is executed to get the answer. For example, in knowledge-based question answering (KBQA) (Berant et al., 2013; Yih et al., 2016; Saha et al., 2018) questions are parsed into SPARQL-like queries consisting of predicates, entities and operators. In Math Word Problem (Huang et al., 2016; Amini et al., 2019) questions are translated to stacks of math operators and quantities.

However, in the real world, many QA tasks cannot be solved by end-to-end neural networks and it is also very difficult to translate questions into any kind of formal representation. Solving Chemical Calculation Problems is such an example. Chemical Calculation Problems cannot be solved by end-to-end neural networks since complex symbolic calculations are required. It is also difficult to translate such problems into formal representations, since not all operators in solving processes occur in question stems, which makes it difficult to annotate data and train models.

Table 1 shows a question in ChemistryQA. To answer the question in Table 1, machines need to: 1) understand the question and extract variable to be solved and conditions in the question; 2) retrieve and apply related chemistry knowledge, including calculating molarity by mole and volume, balancing a chemical equation and calculating the equilibrium constant K, although there is no explicit statement

Table 1: An Example in ChemistryQA.

| Question | At a particular temperature a 2.00 L flask at equilibrium contains $2.80 \times 10^{-4}$ mol $N_2$, $2.50 \times 10^{-5}$ mol $O_2$, and $2.00 \times 10^{-2}$ mol $N_2O$. How would you calculate K at this temperature for the following reaction: $N_2(g) + O_2(g) \to N_2O(g)$ ? [1] |
|---|---|
| Variable to be solved | Equilibrium constant. K of this reaction |
| Conditions provided | Volume of the flask is 2.00L. 
 Mole of $N_2$ is $2.80 \times 10^{-4}$ mol. 
 Mole of $O_2$ is $2.50 \times 10^{-5}$ mol. 
 Mole of $N_2O$ is $2.00 \times 10^{-2}$ mol. 
 Reaction equation is $N_2(g) + O_2(g) \to N_2O(g)$. |
| Knowledge required | K = $\frac{[N_2O]^a}{[N_2]^b[O_2]^c}$, $[*]$ is the molarity of *, and a,b,c are coefficients of matters. 
 Molarity = Mole / Volume. 
 Balance the reaction equation following atomic conservation theory. |
| Solving steps | 1. Balance the reaction equation to get a,b and c. 
 2. Calculate molarities for $N_2$, $O_2$ and $N_2O$ 
 3. Calculate K following K's formula |
| Answer | $4.08 \times 10^8$. |

for "calculating molarity" and "balancing equations" in the question. The combination of these capabilities is scarcely evaluated well by existing QA datasets. In order to foster the research on this area, we create a dataset of chemical calculation problems, namely ChemstriyQA.

We collect about 4,500 chemical calculation problems from `https://socratic.org/chemistry`, covering more than 200 topics in chemistry. Besides the correct answer, we also label the target variable and conditions provided in a question. Such additional labels facilitate potential data augmentation and inferring golden solving process for training.

To verify the dataset is consistent with the purpose of evaluating AI' comprehensive capability and help other researchers ramp up, we build two baselines as follows. a) We build a BERT based sequence to sequence model, which take the raw question as input and the answer as output. The first baseline achieves 0.164 precision on ChemistryQA. b) We create an extraction system which extracts the target variable and conditions from raw questions. The extracted structure information is fed into a graph searching based solver, which performs a sequence of calculating and reasoning to get the final answer. The second baseline achieves 0.169 precision on ChemistryQA.

In summary, our contribution of this paper is shown as follows.

- We propose a new QA task, ChemistryQA, which requires open knowledge and complex solving processes. ChemistryQA is different with other existing QA tasks, and cannot be solved by existing QA methods very well.

- We create a ChemistryQA dataset, which contains about 4,500 chemical calculation problems and covers more than 200 topics in chemistry. In this dataset, we provide a novel annotation for questions, which only labels the variable asked and conditions from question stem but not solving processes. This annotation is much easier and cost less effort, and it is flexible for researchers to explore various of solutions as a weakly supervised dataset.

- We build two baselines to prove: a) end-to-end neural networks cannot solve this task very well; b) the annotation we provided can be used to improve a simple graph search based solver.

## 2 CHEMISTRYQA DATASET

### 2.1 DATA COLLECTION

We collect chemical calculation problems from `https://socratic.org/chemistry`. It this website, there are more than 30,000 questions which cover about 225 chemistry related topics, e.g., Decomposition Reactions, Ideal Gas Law and The Periodic Table. There is an example of annotation page in Appendix A. Figure 2.A shows the original page in Socratic, which contains a raw question,

an answer and probably a description of solving process. We filter raw questions by a simple rule, and only keep questions with a numerical value, a chemical formula or a chemical equation as answers.

## 2.2 DATA ANNOTATION

Unlike similar tasks' annotation, we cannot collect all the atomic operations needed before starting annotation, since the set of chemical operators is not closed. Therefore, we propose a novel annotation method that only the target variable and all conditions will be labeled in a triple-like format. For instance in Figure 2, the target variable is labeled as *(subject = reaction, predicate = Equilibrium constant K, object = ?)*, and one of conditions is labeled as *(subject = $N_2$, predicate = Mole, object = $2.80 \times 10^{-4}$ mol)*.

Therefore, for a question in a link, parts to be annotated are question stems, correct answers, the target variable and all conditions. Figure 2.B shows our annotation page for a question link. For questions and answers, we ask annotators to copy them into corresponding forms. If there are typos or obvious mistakes, we also ask annotators to correct them. For the target variable and conditions, we break them down into several types: *physical unit*, *chemical formula*, *chemical equation*, *substance name* and *other*. We also design easy-to-understand annotation interfaces, e.g., *([BLANK (predicate)] OF [BLANK (subject)] IN [BLANK (unit or None)])* and *([BLANK (predicate)] OF [BLANK (subject)] = [BLANK (object or value)])* for tagging the *physical unit* from the raw question as variables and conditions, respectively. More detail about other types' definitions and annotation interfaces are shown in Appendix A.

We employed crowdsourcing for this annotation work. The task was split into 6 batches and assigned to annotators sequentially. We applied some check-and-verify mechanism in first batch to ensure the annotation quality, also help annotators be more familiar with this task. Finally, we have collected 4418 annotated questions within around 336 hours.

During the annotating phase, we encourage annotators to use text phrase in original questions whenever possible for *chemical formula*, *chemical equation*, *substance name*, subject and value in *physical unit*, while for predicates and units, we do not make any restrictions. We maintain two dynamic mappings to convert mentions labeled to identified predicates or unites, which greatly reduces the difficulty of labeling and the total annotation time. For *other*, there is not any restrictions either, and we only consume identified ones, e.g., STP.

## 2.3 DATA ANALYSIS

We divide annotated questions into train, valid and test subsets, and their sizes are 3433, 485 and 500, respectively. We make some statistics on the annotated questions in different perspectives as follows.

1) According to the types of target variables, we divide questions into 3 classes, *physical unit*, *chemical formula*, *chemical equation*. Table 2 shows examples belonging to different question types, and Table 3 shows the distribution of question types.

Table 2: Examples under various question types

| Question | Answer | Question Type |
|---|---|---|
| How many moles of ammonium nitrate are in 335 mL of 0.425 M NH4NO3? | 0.14 moles | Physical Unit |
| What is the empirical formula of magnesium chloride if 0.96 g of magnesium combines with 2.84 g of chlorine? | MgCl2 | Chemical Formula |
| How would you write a balanced equation for the combustion of octane, C8H18 with oxygen to obtain carbon dioxide and water? | 2C8H18 + 25O2 → 16CO2 + 18H2O | Chemical Equation |

Table 3: Distribution of different question types

| Dataset | Physical Unit | Chemical Formula | Chemical Equation | Total |
|---|---|---|---|---|
| Train | 2,721 | 314 | 398 | 3,433 |
| Valid | 381 | 55 | 49 | 485 |
| Test | 392 | 55 | 53 | 500 |
| Total | 3,494 | 424 | 500 | 4,418 |

2) There are 172 unique predicates, 90 unique units and 25 identified other conditions. We conducted detailed statistics on them in Appendix B,

## 2.4 COMPARING WITH OTHER QA DATASETS

We pick a representative dataset from each type of task to compare with ChemistryQA, including WEBQUESTIONS(Berant et al., 2013), RACE(Lai et al., 2017), ARC(Clark et al., 2018) and MathQA(Amini et al., 2019). We compare these QA datasets in Answer Type, External Knowledge, Knowledge usage, Calculation and Annotation perspectives, and Table 4 shows the detail.

Table 4: ChemistryQA Compares with existing related QA tasks.

| Dataset | Answer Type | Extenal Knowledge | Knowledge usage | Calculation | Annotation |
|---------|-------------|-------------------|-----------------|-------------|------------|
| WEBQUESTIONS | Entity, Entities | Open | Graph Search | SPARQL Operators | SPARQL, Answer |
| RACE | Option | None | None | Language Understanding | Only Answer |
| ARC | Option | Open | Implicit Inference | None | Only Answer |
| MathQA | Option | Closed | Language Understanding | Math Operators | Operator Stack, Answer |
| ChemistryQA | Value, Formula, Equation | Open | Graph Search, Language Understanding and Calculating | Chemical Operators | Variable, Conditions, Answer |

Comparing ChemistryQA with existing QA datasets, ChemistryQA has the following advantages:

1) ChemistryQA contains more diverse answer types and excludes the influence of randomness by not providing options.

2) There are various knowledge required by ChemistryQA including a) triplet-like fact, e.g., substances' molar mass, colour and other physical properties, b) calculation methods between various physical quantities and c) domain specific skills, e.g., balancing chemical equations. The knowledge in ChemsitryQA is open and used in various ways, while other datasets use knowledge in single way.

3) ChemistryQA only provides triplet like extraction annotation which isolates language understanding and domain knowledge as much as possible. This setting makes annotation and model training easier.

## 3 METHODS

We provide two completely different baselines: 1) an end-to-end neural based solver and 2) a solving pipeline composed of an extractor and a graph search based solver.

## 3.1 END TO END SOLVER

We build a sequence to sequence model, and both of its encoder and decoder are based on BERT model. Both encoder and decoder load from pretrained BERT and share the same vocabulary, more than 30,000 sub-tokens from BERT. To build the decoder, we change the encoder's structure as Vaswani et al. (2017) did: 1) the self-attention of decoder has a triangular mask matrix and 2) there is an extra layer of attention performing over outputs of the encoder and hidden states of the decoder. We also append a head of predicting next token to the decoder, which maps hidden states into the vocabulary size space $R^v$ and follows a softmax layer behind it. The end-to-end solver takes the question as the input of encoder and takes the answer as the target of decoder. Questions are split into sub-tokens, and even real numbers also break into sub-tokens. We greedily choose the next token with maximum score after softmax. We append a special token, '[SEP]', as the end of the target sequence. During inference, the decoding process ends when the decoder outputs '[SEP]' token.This method represents a class of powerful neural networks, which achieved state-of-the-art performance on many QA tasks.

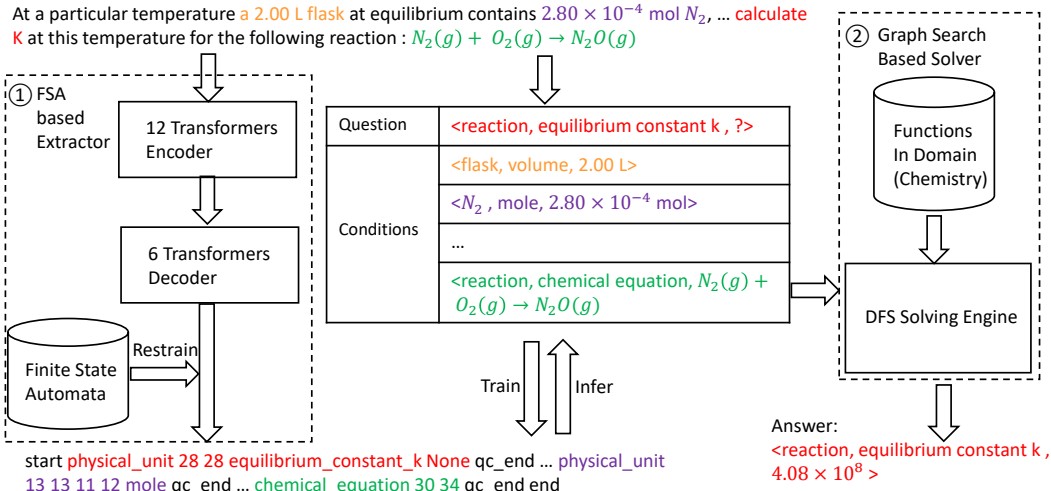

Figure 1: The structure of Extractor plus Graph Search Based Solver pipeline

## 3.2 EXTRACTOR PLUS GRAPH SEARCH BASED SOLVER

As the second baseline, we build an extractor plus solver pipeline. First, we employ the extractor to extract the target variable and conditions from the question text. The target variable and conditions are represented as triplets as described in the above Data Annotation Section. Second, we employ a graph search based solver to take triplets as input and execute pre-defined functions in chemistry domain to get the final answer. Figure 1 shows the structure of the extractor plus solver pipeline.

### 3.2.1 FSA BASED EXTRACTOR

We expect the extractor can take the raw question as input and output triplet like variable and conditions, so a sequence to sequence model is a good candidate. However, such straight forward method is hard to work, because triplets are structural and we also want to extract simple semantic information, i.e., whether a triplet is a target variable or a condition and types of triplets. Therefore, we design a Finite State Automata (FSA) to restrict the type of each output token. We define 12 separate vocabulary sets, and the Table in Appendix C shows the these vocabulary names, sizes and tokens belonging to them. To distinguish between vocabulary and token, we use upper cases to represent vocabulary names and lower cases to represent tokens. START, END, QC_END are control symbols. START and END represent the beginning and end of the decoding sequence, while QC_END represents the end of the target variable or a condition block. PHYSICAL_UNIT, CHEMICAL_EQUATION, CHEMICAL_FORMULA, SUBSTANCE, C_OTHER are types of target variables or conditions. POINTER contains all possible positions of token in question, which can be used to represent the start or end of a text span, e.g., subjects, chemical equations and values.

For the model, we employ a standard pretrained BERT based model as the encoder. We still use 6 layers transformers as the decoder model, and use a triangular mask matrix as the attention mask. We use state transitions to represent the relations between output tokens and build a FSA to model the state transition process. The decoder always outputs "start" as the first token, determines the current state and chooses the next vocabulary based on the current output and state. Figure 1 also shows an example of the output of FSA based Extractor.

In the training stage, we convert triplet-like annotations to the FSA based token sequence, and treat them as the target of the extractor. For example in Figure 1, the target variable, *<reaction, equilibrium constant k , ?>*, is translated to *physical_unit 28 28 equilibrium_constant_k None qc_end*, and one of conditions, *<reaction, chemical equation, $N_2(g) + O_2(g) \rightarrow N_2O(g)$ >* is translated to *chemical_equation 30 34 qc_end*. In the inference stage, we obtain the FSA based token sequence from the decoder, and then we convert it to triplets as the input of subsequent solver. Since there is no conflicts in FSA, both of directed conversions are determinate and lossless.

### 3.2.2 GRAPH SEARCH BASED SOLVER

We get the target variable and all kinds of conditions from the extractor, and then we want to call a sequence of functions to calculate the final answer step by step. However, we do not have such information, because the annotators only label target variables and conditions but not the solving processes. According to our observation, specific physical quantities only can be calculated by functions with specific chemistry knowledge. For example in Figure 1, the equilibrium constant k of the reaction can be calculated by $k = \frac{[N_2O]^2}{[N_2]^2[O_2]}$, where $[]$ is the molarity of a substance. Therefore, we can implement a function, noted as *CalculateK*, which takes the molarity of the three substances and the balanced chemical equation as the input and calculates $k$. However, there is no molarity but only mole of substances extracted, and the chemical equation is not balanced. Therefore, We need to implement a function, *CalculateMolarity*, to obtain molarity of these substances and a function, *BalanceChemicalEquation*, to balance the chemical equation before calling *CalculateK*.

We model this process as a graph search in a hyper graph as follows: 1) We view triplets as nodes in this graph, e.g., *<reaction, chemical equation, $N_2(g) + O_2(g) \rightarrow N_2O(g)$ >*. 2) We view pre-build functions as directed hyper edges, e.g., *CalculateK*, *CalculateMolarity* and *BalanceChemicalEquation*. A hyper edge directs from the input triplets to the output triplets, e.g., the hyper edge, *CalculateK*, starts from $< N_2$,molarity,?>, $< O_2$,molarity,?>, $< N_2O$,molarity,?>,<reaction,chemical equation,$N_2(g) + O_2(g) \rightarrow N_2O(g)$ > and directs to <reaction, k, ?>. 3) The solver maintains a set of triplets without unknown quantities, and uses conditions obtained from the extractor to initialize the set. 4) The solver starts searching from the target variable. If the inputs can be satisfied by the current triplets, then the solver executes the function and adds the result to the set. Otherwise, the solver performs deep searching to check whether the unsatisfied input can be calculated by any function. Table 5 shows the algorithm, where $S$ is the set of triplets without unknown quantities, $I_f$ and $O_f$ are inputs and outputs of function $f$, $I_u$ is a subset of $I_f$ and $\forall i \in I_u : i \notin S$, and $F_p$ is the set of functions with $p$ as their output triplets' predicate.

Table 5: Graph based search solving algorithm

| |
|---|
| **Input**: The target variable $q$, Condition set $C$, Function set $F$ |
| **Output**: Answer $A$ |
| 1: **FUNC** GraphSearch(Target $q= < s, p, t =? >$, Triplet set $S$): |
| 2:    $F_p = \{f | f \in F$ and the predicate of $O_f$ is $p\}$ |
| 3:    for f in $F_p$: |
| 4:        $I_u = \{i | i \in I_f$ and $i \notin S\}$ |
| 5:        for $i$ in $I_u$: |
| 6:            $i$=GraphSearch(i,S) |
| 7:            if $i =< s_i, p_i, t_i >$ is satisfied: |
| 8:                $S = S \cup < s_i, p_i, t_i >$ |
| 9:        if $\forall i \in I_f$ is satisfied: |
| 10:          execute t=f(S) |
| 11:          return $< s, p, t >$ |
| 12:    return None |
| 13:**Call** $A$=GraphSearch($q,C$) |

In Appendix D, we show several functions with inputs and outputs as examples. A triplet with specific predicate can be calculated by different more than one functions with different inputs, e.g., *CalculateMoleFromMass* and *CalculateMoleFromNumber* take mass and the number of atoms(or molecules) as input, respectively. We do not need to implement all functions for all predicate combinations, since there may a combination of functions can represent the relationship among predicates. For example, we can call *CalculateMoleFromMass* and *CalculateMolarity* sequentially to calculate a substance's molarity given its mass. We implemented 78 functions which covers 61 predicates in training set, and the coverage is 35.5%. We list all functions we implemented in Appendix D and will publish them once the paper is published.

## 4 EXPERIMENTS

### 4.1 SETTING

For the end-to-end solver, we use the pretrained uncased BERT base model to initialize its encoder, and use the first 6 layers of BERT to initialize its decoder. We also reuse the tokenizer from BERT for this model, and both encoder and decoder share the sub-token embedding matrix from BERT base.

We tuned the hyper parameters as follows: learning rate $\in \{1 \times 10^{-5}, 5 \times 10^{-5}, 1 \times 10^{-4}, 5 \times 10^{-5}\}$, epoch $\in \{100, 200, 300, 500\}$ and fix batch size as 8.

For the FSA based extractor, we use the same model structure as the end-to-end solver. However, we replace BERT based decoder's vocabulary to 12 customized FSA based vocabularies. We initialize parameters of the extractor by pretrained uncased BERT base model, except the embeddings of tokens from decoder's vocabularies. We randomly initialize the embeddings of tokens from decoder's vocabularies. We also tuned hyper parameters in extractor as exactly same as we did for the end-to-end solver. For the graph search based solver, we restrain the maximum depth as 5 when perform searching. We train both the end-to-end solver and the extractor on a single P40 GPU.

## 4.2 Evaluation and Result

To evaluate the methods, we design different criterion for different types of questions, i.e., physical unit, chemical formula, chemical equation questions. For physical unit questions, if $\frac{|A-\hat{A}|}{A} < 0.05$, we treat the answer is correct. For both chemical formula and chemical equation, we remove spaces in both $A$ and $\hat{A}$ and perform string matching between them. $A$ is the ground true answer and $\hat{A}$ is the predicted answer.

For graph search based solver, we also evaluate the accuracy of the extractor beside the final answer. We evaluate token-level and question-level accuracy for the output sequence from the decoder, respectively calculated by $A_t = \frac{1}{N_q} \sum_i^{N_q} \frac{\sum_j^{Nt_i}(t_j == \hat{t_j})?1:0}{Nt_i}$ and $A_q = \frac{\sum_i^{N_q}(s_i == \hat{s_i})?1:0}{N_q}$, where $t$ and $\hat{t}$ are respectively ground true and predicted tokens, $s$ and $\hat{s}$ are ground true and predicted output sequences, $N_q$ is the number of questions, and $Nt$ is the number of tokens in $s$. $(s == \hat{s})$ is true if and only if $(t == \hat{t})$ is true at all position in $s$.

Table 6: Performances of methods on ChemsitryQA Development Set

| Method | Token-level Accuracy | Seq-level Accuracy | Answer Accuracy |
|---|---|---|---|
| End to End Solver | - | - | 0.164 |
| Extractor + GraphSolver | 0.713 | 0.303 | 0.169 |

Table 6 shows the performances of these two methods. We can obtain the following observations:

a) End-to-end solver achieves 0.164 answer accuracy, which surprises us but also implies that the powerful neural network can learn some pattern of questions, including calculating physical quantities, inferring chemical formulas and equations.

b) The FSA based extractor plus graph search based solver achieves 0.169 answer accuracy even only with 35.5% functions implemented, which implies this framework is effective, and the larger coverage of functions implemented will likely increase the accuracy.

c) For the extractor, the token-level accuracy is 0.713, but the sequence level accuracy drops to only 0.303. This implies the issue of cascading error is serious, but to improve sequence-level accuracy is very difficult. Thus, the more robust subsequent solver is probably needed.

## 4.3 Analysis

We want to analyze reasons from wrong cases and get ratios of them. First, we know there are about 69.7% wrong cases come from the wrong extraction results, and then we sample 100 wrong cases which have the correct extraction results from the development set for analyzing reasons. Table 7 shows the ratios of reasons.

From the analysis, we can observe that most of wrong cases are caused by lacking some chemical knowledge in other forms which cannot be handled by the current solver.

## 5 Related Work

We introduce related work of this paper from two perspectives: 1) There are kinds of examinations employed as benchmark to evaluate machine intelligence. 2) There are some NLP/CV tasks or datasets solved by both end-to-end neural networks and parsing plus execution method.

Table 7: Reasons of wrong cases and some examples

| Reason | Ratio | Examples and Explanation |
|---|---|---|
| Infer Chemical Equation or Formula | 46% | For question "What mass of oxygen is needed for the complete combustion of $8.90 \times 10^{-3}$ g of methane?", the current solver cannot write the correct chemical equation for the combustion reaction. |
| Multiple States | 23% | For questions "How much energy is required to raise the temperature of 3 g of silver from 15 °C to 20 °C?", the current solver cannot tell and maintain more than one different states in the question. |
| Lack of Knowledge | 6% | For question "How many $O_2$ molecules will be consumed when one gram of propane combust completely in the air?", it lacks knowledge about what propane's chemical formula is. |
| Wrong Label | 7% | - |
| Other Reason | 18% | - |

From the perspective of examinations, so far, there have been several tasks proposed based on some question types in specific subjects: 1) Math word problem (Huang et al., 2016; Wang et al., 2017; Amini et al., 2019) can be viewed as a semantic parsing task. Most of them require models translating question texts to equations or sequences of mathematical operations and then performing simple calculations to get the final answer. However, these datasets do not involve domain knowledge and usually provide strong supervision for the parser, i.e., equations or sequences of operations. 2) Newtonian physics datasets (Sachan & Xing, 2018; Sachan et al., 2018) involves physical knowledge, but its knowledge is narrow and limited to Newtonian physics. This dataset is not public. 3) Elementary and middle school science examination (Clark, 2015; Clark et al., 2018) contains multiple choice questions involving open knowledge crossing various subjects. However, many questions in the dataset can be solved by retrieval and text matching. Although ARC(Clark et al., 2018) separates a subset of hard questions, the state-of-the-art on ARC is retrieval plus a powerful transformer based model. 3) Biological processes problem dataset (Berant et al., 2014) provides passages to describe biological processes and asks questions about the passages. Different from ours, this dataset focuses more on evaluating machine reading comprehension, as English reading comprehension data (Lai et al., 2017) does.

From the perspective of solving methods, besides examinations, there are several datasets can be solved by both end-to-end models and parsing plus execution method. For example, WEBQUES-TIONS(Berant et al., 2013) is a famous KBQA dataset, and there are both end-to-end models (Bordes et al., 2014; Hao et al., 2017) and semantic parser (Yih et al., 2014; Berant & Liang, 2014) working on it. For WEBQUESTIONS, the solving process (i.e., executing SPARQL on Freebase) is fix after obtaining the parsing result, and the correct parsing result must lead to the correct answer. However, for ChemistryQA, there is more than one paths from extraction result to the correct answer, and it requires searching on the graph. Another example is Knowledge Base Completion (KBC) task. KBC task can be solved by both end-to-end knowledge graph embedding models (Bordes et al., 2013) and logical inference based method, e.g., Markov Logic Network (Wei et al., 2015). However, the input of KBC is not natural language.

## 6 CONCLUSION

Real world question answering is more complex than existing QA tasks, since it requires not only understanding questions well but also interleaving complex reasoning with knowledge retrieval, which is scarcely represented by existing QA datasets. To foster the research on this area, we create ChemstriyQA dataset which contains chemical calculation problems. We implemented two baselines, a sequence to sequence model and a FSA based extractor plus a graph search based solver, which stands for two types of methods, the end to end neural network and the extractor plus solver, respectively. The experiment result shows the extractor-plus-solver baseline can achieve a better performance with only 35.5% functions in domain implemented. Therefore, there is still room for improving in the extractor-plus-solver method, but it is hard to improve performance for the end to end models.

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

# A    ANNOTATION EXAMPLES, INTERFACES AND PROCESSES

## A.1    AN EXAMPLE OF ANNOTATION PAGE

Figure 2 shows a page of annotation, the left part is the original web page in `socratic.org` and the right part is the annotation area.

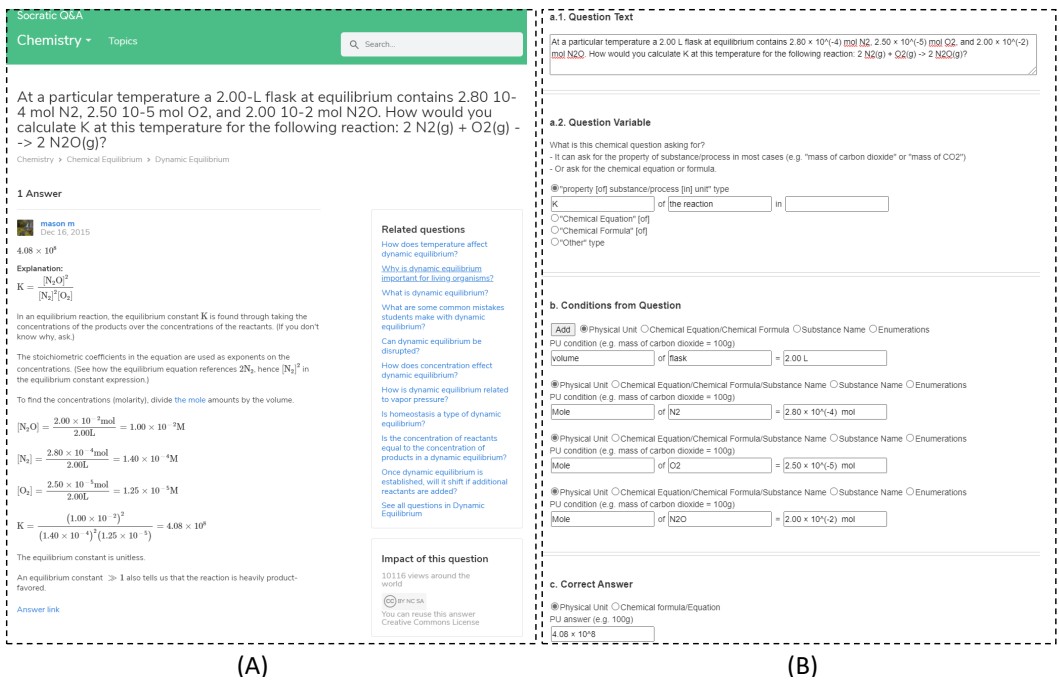

Figure 2: An example of the annotation page

## A.2    ANNOTATION TYPES

The annotation area contains two parts: one for labeling the target variable and the other one for labeling conditions. For a chemical calculation problem, there is only one target variable but probably more than one conditions. Thus, annotators are free to add condition block. Usually, the interface for question variables and conditions are different even for the same type of annotations. For a variable or condition block will be first ask to choose a type of annotation, and then the page will show the interface of it. Table 8 shows interfaces for all types of annotations.

Table 8: Annotation Interfaces

| Annotation Type | As a Question Variable | As a Condition |
|---|---|---|
| physical unit | [BLANK (predicate)] OF [BLANK (subject)] IN [BLANK (unit or None)] | [BLANK (predicate)] OF [BLANK (subject)] = [BLANK (object or value)] |
| chemical formula | Chemical Formula OF [BLANK (subject)] IN NONE | Chemical Formula OF [BLANK (subject)] IS [BLANK (value)] |
| chemical equation | Chemical Equation OF reaction | Chemical Equation OF reaction IS [BLANK (value)] |
| substance | - | Substance IS [BLANK (value)] |
| other | - | Other Condition IS [BLANK (value)] |

### A.3 CROWD SOURCING ANNOTATION DETAILS

First, we estimated the annotation time by recording time spent on annotators label a small scale experimental set, and got the average time spent on labeling and verifying are 4.65 minutes and 3.25 minutes, respectively. In earlier stage, we performed both labeling and verifying on each question. After we got the verified rate is high enough, we only keep the labeling process, which leads to the annotation time of a question is 4.4 minutes. Finally, we obtain 4418 annotated questions with spending about 336 hours.

## B PREDICATES, UNITS AND OTHERS IN CHEMISTRYQA

Table 9 shows top predicates and units.

Table 9: Statistics on top predicates and units

| Predicate | Count | Unit | Count |
|---|---|---|---|
| mass | 2,013 | gram | 1,911 |
| volume | 1,851 | mol/l | 1,098 |
| temperature | 1,099 | mol | 1,033 |
| mole | 1,056 | liter | 899 |
| molarity | 731 | °C | 872 |
| pressure | 709 | mliter | 832 |
| concentration | 323 | atm | 398 |
| ph | 226 | kelvin | 222 |
| number | 226 | mmhg | 168 |
| heat energy | 146 | kpa | 164 |

We list all predicates in Table 10, list all units in Table 11 and list all other conditions in Table 12.

Table 10: All predicates in ChemsitryQA

| [ch3cooh] | [h+] | [h2so4] | [h3o+] | [nh3] |
|---|---|---|---|---|
| [nh4+] | [oh-] | absorbance | absorbance reading | absorptivity |
| acid concentration | activation barrier | actual yield | altitude | angle of the reflected beam |
| area | atomic mass | average oxidation number | avogadro constant | barometric pressure |
| boiling point temperature | cell potential | changed factor | charge | coefficient |
| completion percent | composition by mass | concentration | constant r | coordination number |
| cp | degree of dissociation | delta vaph | deltag | $deltag^0$ |
| deltagf | deltah | deltah(rxn) | $deltah^0$ | $deltah^0f$ |
| $deltahf^0$ | deltas | density | depth | diameter |
| dilution factor | distance | ecell | electric current | electron configuration |
| energy | energy transfer rate | enthalpy | enthalpy of combustion | enthalpy of formation |
| enthalpy of fusion | enthalpy of sublimation | enthalpy of vaporization | entropy | equilibrium concentration |
| equilibrium constant k | equilibrium constant ktant | equivalent weight | first ionisation energy | formula weight |
| freezing point temperature | gauge pressure | half-life | hardness | heat capacity |
| heat capacity ratio | heat energy | height | initial temperature | ionic product |
| ionization energy | ka | kb | kbp | kf |
| kh | kinetic energy | kp | kw | longest wavelength |
| mass | mass/volume percent | mass concentration | mass percent | maximum amount |
| maximum possible yield | melted time | melting point temperature | molal solubility | molality |
| molar enthalpy of dissolution | molar enthalpy of vaporization | molar heat | molar mass | molar ratio |
| molar solubility | molar volume | molarity | molarity percent | mole |
| mole fraction | molecular weight | multiplier | normality | number |
| o.n | osmolarity | osmotic pressure | oxidation number | oxidation state |
| partial pressure | path length | percent | percent composition | percent ionization |
| percent yield | ph | pka | pkb | poh |
| power | pre-exponential factor | pressure | quantity | radius |
| rate law | ratio | reaction enthalpy | reaction quotient q | reaction rate constant |
| relative molecular weight | rms speed | size | solubility | solubility product |
| specific heat | specific heat capacity | standard atmospheric pressure | standard emf | standard enthalpy |
| standard enthalpy change | standard enthalpy formation | standard free energy | state n | steric number |
| strength | surface tension | temperature | theoretical yield | thermal energy |
| thermochemical equation | threshold odor number | time | times | total pressure |
| trasmittance | value | van t hoff factor | vapor pressure | vapour density |
| velocity | voltage | volume | volume percent | w/w |
| wavelength | | | | |

Table 11: All units in ChemsitryQA

| | | | | |
|---|---|---|---|---|
| $(M * min)^{(-1)}$ | amperes | amu | atm | bar |
| cal | cal/(g · °c) | cal/mol | cm/s | $cm^3$ |
| $dm^{(-3)}$ | $dm^3$ | eq/l | fluid ounces | g |
| g/100 ml | $g/cm^3$ | $g/dm^3$ | g/eq | g/l |
| g/ml | g/mol | gallon | hour | $in^{(-2)}$ |
| $in^2$ | j | j/(g · k) | j/(g · °c) | j/(g · °c |
| j/(kg · k) | j/(kg · °c) | j/(mol · k) | j/(mol · °c) | j/(°c · g) |
| j/g | j/mol | k | kcal | kcal/g |
| kcal/mol | kg | kj | kj/(kg · °c) | kj/min |
| kj/mol | kpa | l | l/(mol · s) | m |
| m/s | $m^3$ | meq/l | mg | mg/l |
| mg/ml | milliequivalents | min | mj | ml |
| mm | mmhg | mmol | mol | mol/(l · s) |
| $mol/dm^3$ | mol/g | mol/kg | mol/l | mpa |
| n | ng | nm | none | osmol |
| osmol/l | oz | pa | particles/s | pounds |
| ppm | psi | s | $s^{(-1)}$ | t |
| torr | v | $years^{(-1)}$ | °c | °c/(kg · mol) |
| l | | | | |

Table 12: All other conditions in ChemsitryQA

| | | | | |
|---|---|---|---|---|
| Avogadro constant | Combusted | Conjugate base | vaporize | Equilibrium |
| Equivalence | Freezing | Henderson-Hasselbalch | Incomplete combustion | NTP |
| Neutralization | OTHER | Photosynthesis | SATP | STP |
| Stoichiometrically | burn | combustion | complete combustion | constant temperature |
| decomposes | ionizing | melt | neutralize | reaction |
| ConstantTemperaturePressue | | | | |

## C    FSA VOCABULARIES AND STATE TRANSITIONS

Table 13 shows the vocabularies used in Finite State Automata of Extractor and some tokens in them.

Table 13: The vocabularies' names and tokens belonging to them.

| Vocabulary | Size | Example Tokens |
|---|---|---|
| START | 1 | ["start"] |
| END | 1 | ["end"] |
| QC_END | 1 | ["qc_end"] |
| POINTER | 512 | ["0","1","2","3","4","5","6","7","8","9","10",...,"509","510","511"] |
| PHYSICAL_UNIT | 1 | ["physical_unit"] |
| CHEMICAL_EQUATION | 1 | ["chemical_equation"] |
| CHEMICAL_FORMULA | 1 | ["chemical_formula"] |
| SUBSTANCE | 1 | ["substance"] |
| C_OTHER | 1 | ["c_other"] |
| PREDICATE | 172 | ["atomic_mass","boiling_point_temperature","energy", "concentration","heat_capacity", "molarity", ...] |
| UNIT | 91 | ["atm","bar","cal/(g_·_°c)","j/(g_·_k)","min","mol","ppm","psi",...] |
| OTHER | 26 | ["Avogadro_constant","Combusted","Conjugate_base",...] |

We define 26 states for FSA. Table 14 shows all states and Figure 3 shows the state transitions.

Table 14: States of FSA

| s_start | s_q_pu | s_q_pu_subject_start | s_q_pu_subject_end | s_q_pu_predicate |
|---|---|---|---|---|
| s_q_pu_unit | s_c_pu | s_c_pu_subject_start | s_c_pu_subject_end | s_c_pu_value_start |
| s_c_pu_value_end | s_c_pu_property | s_q_ce | s_q_cf | s_q_end |
| s_c_ce | s_c_cf | s_c_sub | s_c_other | s_c_ce_start |
| s_c_ce_end | s_c_cf_start | s_c_cf_end | s_c_sub_start | s_c_sub_end |
| s_c_other_type | | | | |

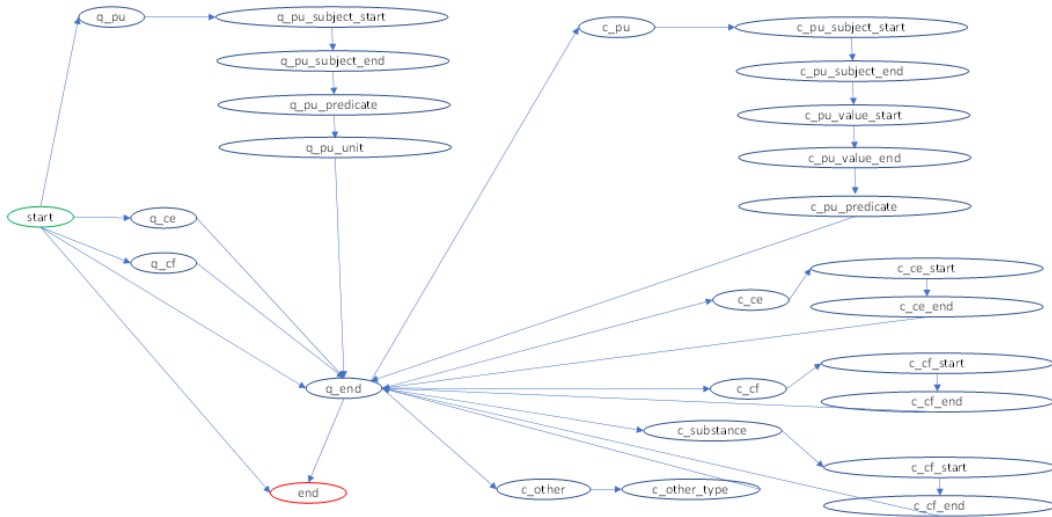

Figure 3: The state transition graph for the FSA

## D    FUNCTIONS

Table 15 shows several functions' inputs and outputs for example.

Table 15: Function examples implemented for ChemistryQA

| Function Name | Input Predicates | Output Predicates |
|---|---|---|
| *CalculateK* | balanced chemical equation, molarity (for all substance in equation) | equilibrium constant k |
| *CalculateMolarity* | mole, volume | molarity |
| *CalculateMoleFromMass* | mass, molar mass | mole |
| *CalulcateMoleFromNumber* | number of atoms or molecules | mole |
| *BalanceChemicalEquation* | chemical equation | chemical equation |
| *ParseChemicalEquation* | text span | chemical equation |

We also list all functions we implemented as follows:

FFunc_Name2CE

Func_Formula2CE

Func_Equation2CE

Func_BalanceChemicalEquation

Func_Mole2Atom

Func_CE2MolarMass

Func_Mass2Mole

Func_Ph2Kw

Func_Number2Mole

Func_Mole2Number

Func_MassMolar_mass2Mole

Func_MoleMolar_mass2Mass

Func_VolumeMolarity2Mole

Func_VolumeMoleTemperature2Pressure

Func_VolumeTemperaturePressure2Mole

Func_PressureMolar_massTemperature2Density

Func_PressureDensityTemperature2Molar_mass

Func_MoleMass2Molar_mass

Func_MoleVolumePressure2Temperature

Func_MoleTemperaturePressure2Volume

Func_MoleVolume2Molarity

Func_MoleVolume2Concentration

Func_MolarityVolume2Mole

Func_MoleMolarity2Volume

Func_Ph2Acid_concentration

Func_Acid_concentration2Ph

Func_Ph2Poh

Func_Poh2Ph

Func_Ka2Pka

Func_Mass_concentrationMolar_mass2Molarity

Func_MassVolume2Density

Func_DensityVolume2Mass

Func_MassDensity2Volume

Func_MolarityMolar_mass2Molality

Func_MolalityMolar_mass2Molarity

Func_MolarityTemperature2Osmolarity

Func_Theoretical_yieldPercent_yield2Actual_yield

Func_Actual_yieldPercent_yield2Theoretical_yield

Func_Theoretical_yieldActual_yield2Percent_yield

Func_Ka2Degree_of_dissociation

Func_2Freezing_point_temperature

Func_Gauge_pressure2Pressure

Func_MassVelocity2Kinetic_energy

Func_KaMolarity2Percent_ionization

Func_2Standard_atmospheric_pressure

Func_MolalityMolar_mass2Ww

Func_WwMolar_mass2Molality

Func_MassMass2Mass_percent

Func_MoleMole2Mole_percent

Func_MolarityMolarity2Molarity_percent

Func_Poh2Alkali_concentration

Func_AtomMoleculeMole2Mole

Func_MassSpecific_heat_capacityTemperature2Heat_energy

Func_Heat_energySpecific_heat_capacityTemperature2Mass

Func_Heat_energyTemperatureMass2Specific_heat_capacity

Func_Heat_energySpecific_heat_capacityMass2Temperature

Func_MassVolume2Mass_concentration

Func_MassVolume2Density

Func_BptHvapPressurePressure2Bpt

Func_Molarity3_2Ka

Func_Molarity3_2Kb

Func_MolarityDepthAbsorbance2Absorptivity

Func_AbsorbanceMolarityAbsorptivity2Depth

Func_MolarityDepthAbsorptivity2Absorbance

Func_Absorbance2Transmittance

Func_Volume2_2Dilution_factor

Func_ResistanceVoltage2Electric_current

Func_Electric_currentVoltage2Resistance

Func_Electric_currentResistance2Voltage

Func_DensityHeight2Gauge_pressure

Func_Mass2_Time2HalfLife

Func_Heat_of_fusionMolePower2Melted_time

Func_ShcMolar_mass2Molar_heat

Func_MoleculeMolarity2Ph

Func_Chemistry_Equation2K

Func_GetCoefficient

Func_Chemistry_Equation2K

Func_Chemistry_Formula2Oxidation_number

