# OpenReview forum: "ChemistryQA: A Complex Question Answering Dataset from Chemistry"
_ICLR.cc/2021/Conference — Reject_

### Official Review · AnonReviewer2 · 2020-10-25
**Official Blind Review #2**

**Rating:** 5
**Confidence:** 3

**Review:**

Paper Summary:
* This paper presents a question answering dataset called ChemistryQA.  It is different from existing datasets in that ChemistryQA requires open knowledge and complex solving processes.  It provides triplet like extraction annotation which isolates language understanding and domain knowledge. Experimental results show that a neural encoder-decoder model and an extractor-plus-solver do not work well.

Strengths:
* The dataset contains real-world QA that requires the ability to perform complex chemistry calculation and reasoning. It is difficult for crowdsourcing workers to generate such complex questions.
* The authors proposed a novel annotation method that target variables and conditions are labeled in a triple-like format.

Weaknesses:
* The dataset seems small to acquire the ability to perform complex calculation and reasoning. The training, validation, and testing datasets consist of 3,433, 485, and 500 questions, respectively.
* The paper does not show statistics of the dataset such as the average length of questions and answers and the unique number of answers.
* The paper does not show the performances broken down by question types. Although the end-to-end solver achieves an answer accuracy of 0.164, I think it is important to show more detail on what it can and cannot do.
* The authors uses a pre-trained BERT as the encoder of the end-to-end solver and trained the decoder from scratch.  I think pre-trained encoder-decoder models such as T5 and BART are better as the baselines of the end-to-end solver than the model used in this paper.

Review Summary:
* The paper is well motivated. ChemistryQA can be a useful dataset to evaluate the ability of chemistry calculation and reasoning, while the dataset seems small to acquire the ability.   I think it can benefit a lot with a more comprehensive analysis of evaluation results of baselines.

---

### Official Review · AnonReviewer1 · 2020-10-25
**Lofty motivations and goals, but final dataset may not meaningfully accelerate progress toward such goals.**

**Rating:** 3
**Confidence:** 4

**Review:**

*Summary*: This paper proposes a new dataset based on textbook / classroom chemistry questions for complex knowledge retrieval and aggregation. The authors scrape several thousands questions from online repositories and add additional natural language annotations signifying the quantities to be solved for in each question, as well as the declarative knowledge. Two baselines, one end-to-end neural and another symbolic, both fail at this dataset.

*Strengths*: The dataset targets the important question of how to build models that can retrieve knowledge while performing complex reasoning.

*Weaknesses*: As-is, the dataset fails to target the knowledge retrieval component---models are either expected to magically know how to calculate the answer, or use hard-coded functions that complete a graph of values. The neural baseline also seems a bit non-standard, raising questions of how well modern systems can actually do on the task; furthermore, the end-to-end neural system is disadvantaged in that it likely has not seen much chemistry-related content during fine-tuning, whereas the symbolic baseline has access to a host of human-defined functions. Furthermore, dataset quality is a bit difficult to assess without more samples.

*Recommendation*: 3 . This benchmark is motivated by the lofty goals of encouraging the development of models that can combine knowledge retrieval, complex reasoning, and language understanding. However, it’s unclear to this reviewer whether it will prove useful in making progress towards such goals---they’re too conflated to be meaningfully evaluated within this context. To improve the benchmark and make it more amenable toward advancing those research goals (versus just being a difficult datasets that current models cannot handle), I’d recommend explicitly targeting and evaluating this knowledge retrieval component as well. For instance, given a specific knowledge-base that’s guaranteed to span the facts necessary to answer the questions, how well can a model (1) retrieve relevant information and (2) use such relevant information to answer questions?

Questions:

“Chemical Calculation Problems cannot be solved by end-to-end neural networks since complex symbolic calculations are required”: this is a hypothesis---there are many tasks where “complex symbolic calculations are required”, but end-to-end networks excel.

What extent of knowledge is required to solve this task? For instance, many old semantic parsing datasets came with databases, and it was guaranteed that within the database, an answer would occur. What would a corresponding knowledge graph for this case look like, and how complex would it be?

“Unlike similar tasks’ annotation, we cannot collect all the atomic operations needed before starting annotation, since the set of chemical operators is not closed.” The set of mathematical operators is also not closed (e.g., in math word problems). Why is this approach better than collecting all the operations represented in the dataset (even if it doesn’t cover all of the operations that one could conceivably see)?

The annotation interface / process looks quite regular---you aren’t expecting too much variation from the templates given. Given that you can help crowdworkers with these templates, why not just use these templates as the baseline for a formal meaning representation that would encompass the knowledge needed for the task?

Can you give more details about the annotation process, beyond the short paragraph near the end of section 2.2? (“We employed crowdsourcing for this annotation work...around 336 hours)”. I’d be surprised if any crowdworker could label this sort of data well. What quality control filters did you put in place? Can we see more (random) samples of the dataset, so we can better assess its quality?

End-To-End Solver: where did you get this model architecture from, such that “This method represents a class of powerful neural networks, which achieved state-of-the-art performance on many QA tasks.”? I’ve never seen BERT used in a seq2seq setting like this (instead, people tend to use models trained on input/output pairs, like BART or T5). I’d like to see how this compares to using BART or T5, since it’s not clear that the BERT initialization would be good for generation.

Graph-Search based Solver: the need to implement specific functions (78, in this case) is significant, and undermines the point of this dataset, in my opinion. There’s no inherent value in learning to solve chemical equations well---the hope is that, in the process of doing so, we’ll get modeling insights into what methods work well and can be generally applied to other knowledge-intensive tasks. This graph-search based solver seems narrowly scoped to ChemistryQA and difficult to adapt to other tasks, and it’s not entirely clear why we should value its results.

Token-level accuracy: Is it guaranteed that the output of the graph-search based solver will be the same length as the gold output? How? Else, how is token-level accuracy computed?

---

### Official Review · AnonReviewer3 · 2020-10-28
**Summary:  The paper propose a new complex dataset for QA, Chemistry QA, for the task of answering chemistry questions requiring mathematical and symbolic reasoning. They also provide two simple baselines. They claim that end-to-end neural networks cannot solve this task easily and efficiently.**

**Rating:** 5
**Confidence:** 4

**Review:**

Strengths:  A new QA dataset for chemistry QA consisting of 4500 questions and covering 200 topics. Crowdsourced annotation of variable and conditions.

Weaknesses: -  Strong baseline results are missing. Intermediate steps is missing in the annotations which are really helpful in training an end-to-end model.  The topic distribution is missing from the paper. Experimental results and analysis is not enough.

Overall: The idea of curating and annotating a new dataset for Chemistry QA dataset is good. I feel a stronger baseline would have helped much in understanding and analysing the quality dataset and annotations. Also the question complexity analysis/topic distribution is missing. Overall the paper writing could be improved a lot, in the current version it is difficult to follow.

Question: If the whole problem can be converted into a set of triples and conditions they why not use graph based QA techniques?  It will be interesting to see how neural symbolic machines/Neural module network perform on this dataset ? Topic distribution, question type distribution etc are missing.  Any specific reasons for using 12 layers of transformer encoders and 6 layers of transformer decoders in Extractor plus Graph Search Based Solver?  Which graph search algorithm is used in section 3.2.2 and Table 5?


Typos:
 It this website,  -> On this website

---

### Official Review · AnonReviewer4 · 2020-10-28
**Interesting dataset, but evaluation can be improved**

**Rating:** 4
**Confidence:** 4

**Review:**

This paper proposes a new dataset for the chemistry domain as a real-world QA task.
The authors collected chemistry questions from a web page and used crowdsourcing to annotate the questions with labels and conditions required for solving them.
As baseline models, the authors propose an end-to-end neural network model and a symbolic solver that uses pre-defined rules. They demonstrate that the neural model struggles to solve the task but the symbolic solver outperforms it with the pre-defined rules that cover only a part of predicates in the dataset, arguing that their dataset is challenging and can foster the research of real-world problems in the chemical domain.

Pros:

- The paper proposes a new dataset, ChemistryQA, that consists of 4,500 questions with labels of target variables and conditions provided in the questions.
- The paper proposes a graph-search based solver with an extraction mechanism which is carefully implemented for this task.
- Appendix has thorough lists of the predicates and units in the dataset and functions used in the baseline solver.

Cons:

- Overall, I think that the writing of the paper can be improved more. There are some typos and formatting issues, which reduce the paper strength. Besides in Sections 2.1, 2.2, and 3.2.1, the paragraphs refer to figures and tables in Appendix. This seems to violate the submission instructions.
- My primary concern is on the quality of collected questions. In Section 2.2, the authors say that they performed some check-and-verify mechanism in the first batch, which should be described in detail. Some related questions:
  + Did the author compute the inter-annotator agreement on sampled questions?
  + What kind of rules did you define for the verification?
  + How many workers did work on the annotation in total?
  + Did the same pool of workers work on the annotation for the first batch and the subsequent batches?
  + Is there actual human performance on the collected questions? Seemingly, it is not guaranteed that posted questions on the web page are reasonably answerable.
- The purpose of employing two baseline models is not explained well. In the introduction, the authors say that "to verify the dataset is consistent with the purpose of evaluating AI' comprehensive capability". However, their hypotheses for the experiments are not clearly stated.
- It seems that the authors stopped implementing the predefined functions for the symbolic solver at the point where the solver outperforms the neural network model. The authors could have implemented more, but it is not clearly explained why they only implemented the predefined functions for the 35.5% predicate coverage. What would happen if there are a larger number of functions implemented?
- Comparison with existing datasets can be elaborated more. For example, because the Option answer type should include different types of entities and the option is just a form, directly comparing it with value, formula, and equation does not make sense to me.
- In the error analysis, 18% of cases are classified into Other Reason, which does not look like a small number to me. Can the authors break this down into more detailed categories?

Typos:

- Section 2.1, It this website -> In this website
- Cite Devlin et al. (2019) for using BERT.
- There is an ill-formatted cell in Table 4.
- There is inconsistent use of decimal commas (4,500 vs 4418)
- Add whitespace before citations (e.g., in Section 2.4).

---

### Decision · Program_Chairs · 2021-01-07
**Final Decision**

**Decision:**

Reject

**Comment:**

The authors propose a new dataset, ChemistryQA which has complex questions requiring scientific and mathematical reasoning. They show that existing SOTA models do not perform well on this dataset thereby establishing the complexity of the dataset.

The reviewers raised several concerns as summarised below:

1) Writing is not very clear
2) The quality of the dataset is hard to judge as some crucial information about the dataset creation process is missing
3) The size of the dataset is small
4) some stronger QA baselines need to be included

Unfortunately the authors did not provide a rebuttal. Hence, its current form this paper cannot be accepted.